# Effects of Diathermy on Pain in Women with Fibromyalgia: A Randomized Controlled Trial

**DOI:** 10.3390/biomedicines12071465

**Published:** 2024-07-02

**Authors:** Edurne Úbeda-D’Ocasar, Daniela González-Gerstner, Eduardo Cimadevilla-Fernández-Pola, Cristina Ojedo-Martín, Juan Hernández-Lougedo, Juan Pablo Hervás-Pérez

**Affiliations:** 1Faculty of Health Sciences-HM Hospitals, University Camilo José Cela, Urb. Villafranca del Castillo, 49, 28692 Madrid, Spain; eubeda@ucjc.edu (E.Ú.-D.); daniela.gonzalez@alumni.ucjc.edu (D.G.-G.); ecimadevilla@ucjc.edu (E.C.-F.-P.); cojedo@ucjc.edu (C.O.-M.); jlougedo@ucjc.edu (J.H.-L.); 2Instituto de Investigación Sanitaria HM Hospitales, 28015 Madrid, Spain

**Keywords:** fibromyalgia, diathermy, pain, quality of life, chronic fatigue, anxiety

## Abstract

(1) Background: The main characteristic of fibromyalgia (FM) is generalized musculoskeletal pain. This may be accompanied by muscle and joint stiffness, sleep and mood disorders, anxiety and depression, cognitive dysfunction, and chronic fatigue. It is endemic in developed countries, with a higher prevalence among women than men, and its etiology is still unknown. Diagnosis is made based on chronic generalized pain and through the presence of tender points. The objective of this study was to analyze the efficacy of diathermy on pain in patients with fibromyalgia. (2) Methods: A single, blind, randomized experimental study was developed with a sample of 31 participants. Measurements were taken and recorded at three different intervals using the following measurement tools: the pressure pain threshold (PPT) at the tender points (TP) of the right and left trochanteric prominence with an algometer, the pain measurement scale, the Fibromyalgia Impact Questionnaire, the sleep quality index (PSQI, Pittsburgh), the Multidimensional Fatigue Inventory (MFI-S), and the scale for anxiety and depression (Hospital Anxiety and Depression Scale). Sociodemographic data were collected through Google Forms (age, height, weight, Body Mass Index). The intervention took place twice weekly across four weeks of sessions. (3) Results: Statistically significant results were obtained in the right and left trochanter PPT, as well as for anxiety and fatigue in the experimental group. The results obtained show that this treatment has managed to improve the quality of sleep, the impact of disease, chronic fatigue, and anxiety in patients with FM. (4) Conclusions: Diathermy is a tool that can help reduce pain. It can also improve the baseline levels of chronic fatigue, anxiety, the impact of the disease, and sleep quality in patients with fibromyalgia.

## 1. Introduction

Fibromyalgia (FM) is a syndrome characterized mainly by persistent, widespread musculoskeletal pain. The main symptoms are muscle and joint stiffness, insomnia, chronic fatigue, intestinal disorders, mood disorders, cognitive dysfunction, anxiety, depression, general sensitivity, and an inability to carry out daily activities and routines [1]. Additionally, individuals with this condition may experience headaches, irritability, and heightened sensitivity to temperature changes [2,3].

This disease can be related to infections, diabetes, rheumatic disease, and psychiatric or neurological disorders [1,2]. FM is a syndrome with a serious impact on the quality of life of those who suffer from it [3,4,5]. People diagnosed with fibromyalgia syndrome (FMS) often perceive uncertainty as a danger to their quality of life [6].

This syndrome is endemic in developed countries, where it has a prevalence rate of 2.1%, and has increased notably in recent years [3,7]. It is estimated that between 2 and 8% of the world’s population suffers from this pathology [8], affecting 4.1% of Spanish women (20 women for every man) [9]. Various factors have been linked to this observation, including elevated levels of anxiety and depression, as well as pain and hormonal influences [1]. The age range in which FM usually appears is between 30 and 35 years old [10]. Pharmacotherapy, the most common treatment for the disease, can reduce pain by 25 to 40% [11,12,13].

The etiology and pathophysiology of FM are still not well understood [14,15,16,17]. While the precise cause of FM is uncertain and includes unexplained aspects, it is primarily attributed to central sensitization (CS), which involves dysfunctions in the neurological pathways that are responsible for perceiving, transmitting, and processing nociceptive stimuli, with manifestations of generalized musculoskeletal pain [1]. 

Central sensitization is defined as an augmentation of responsiveness of central neurons to input from unimodal and polymodal receptors [18] and encompasses altered sensory processing in the brain, the increased activity of pain facilitatory pathways, and temporal summation of second pain or wind-up [19]. CS contributes to the amplification of neuronal signals in the central nervous system, resulting in an enhanced perception of pain. As a result, patients may experience increased receptive pain areas, allodynia, and hyperalgesia [20].

The pathophysiology of FM has several hypotheses: FM is associated with hypo-reactivity due to the saturation of the hypothalamic–pituitary–adrenal axis, which influences stress, metabolism, and the immune system [3,21]. Within the cortex of the adrenal gland, cortisol hormone production occurs and is associated with stress levels and blood glucose concentrations. In some investigations, it has been shown that a higher concentration of cortisol is related to a lower intensity of pain; consequently, low concentrations of cortisol are related to higher levels of pain [22,23]. One of the hypotheses suggested is that the excessive sensory innervation of the glabrous skin, at the level of arteriole–venule shunts, could lead to irregular blood flow and may play a role in causing severe pain and widespread fatigue in FM patients [12]. This can be observed in 49–63% of patients with FM, as well as sensory–motor axonal and/or the demyelinating polyneuropathy of large fibers, which occurs in 90% of patients [3]. 

To date, there are no specific tests to diagnose FM. Up to 75% of people with the condition remain undiagnosed [24]. In 1990, the committee of the American College of Rheumatology (ACR) drew up diagnostic criteria [25]. FM was diagnosed based on the presence of a pain response when applying approximately 4 kg/cm^2^ in at least 11 out of 18 tender points (TPs) in musculoskeletal structures and chronic widespread pain with 88.4% sensitivity and 81.1% specificity [26]. Subsequently, in 2010, the ACR updated the criteria, including the presence of bilateral generalized pain (above and below the waist, and axial skeletal pain directly related to TP), the presence of somatic symptoms for at least 3 months, and where any other pain-causing disorder has been excluded [27,28]. In a study where pain pressure and subjective pain perception thresholds were evaluated, it was concluded that the most painful point when applying normalized pressure was in the trochanteric prominence [29].

Diathermy (DT) is a non-invasive pain therapy based on the local application of high-frequency electromagnetic waves [30]. This procedure produces deep heat, promotes tissue repair, and influences pain sensitivity [31]. In addition, it allows high-frequency energy to be transferred to deeper levels without heating either the skin/tissue surface to be treated or the metallic surface of the treatment device [32,33]. At the same time, it allows for a greater degree of focus and density of the applied energy. It may act as analgesic therapy for patients with chronic pain [3,34,35,36].

Among the physical properties of DT is the increase in body temperature. It can heat tissues at a deeper level than any other superficial instrument in a shorter application time [37]. The increase in body temperature is related to a decrease in the state of anxiety, as well as an increase in the subjective evaluation of the quality of life [38,39]. The main objective of this study is to investigate if the application of DT in patients with FM has significant positive effects in terms of pain reduction when measured according to VAS and algometry to determine if it can be an indicated technique for the treatment of this syndrome.

We aimed to observe the effect of DT on chronic fatigue, pain, anxiety, and depression; analyze whether there are improvements in the impact of the disease in patients with FM after the application of DT; and check if there is an improvement in the quality of sleep after the application of DT. 

## 2. Materials and Methods

### 2.1. Type of Study

An experimental study design was carried out through a single-blind, randomized study. Patients were randomly assigned to either the experimental or the control group. All the participants and the assessor who measured the variables were blinded.

This study was approved by the Ethics Committee of the University Camilo José Cela (May 2022, code 05_22_SCYFMEDD). 

### 2.2. Sample Characteristics and Inclusion Criteria 

The study involved 31 participants with FM, ranging in age from 30 to 70. Some patients had coexisting diseases that did not prevent the application of DT. The patients were recruited through the Association of Fibromyalgia and Chronic Fatigue of Madrid “Afinsyfacro” (Móstoles, Madrid, Spain). Based on our previous study [40], a minimum sample size of at least 30 patients with FMS was considered appropriate for this study.

The inclusion criteria for participants required them to experience chronic generalized pain and be diagnosed with FM for at least three months. Exclusion criteria included recent surgery, skin conditions unsuitable for DT application, the presence of certain neuropathic conditions (lupus, rheumatoid arthritis, diabetic polyneuropathy), ongoing pharmacological treatment such as anticoagulants within three days before participation, any underlying medical conditions like fractures or tumors; cardiac pathologies including heart failure, uncontrolled arterial hypertension, arrhythmias, and phlebitis thrombi arteriopathies; and having a pacemaker or suffering from epilepsy. 

### 2.3. Procedure 

Once the sample was collected, the patients received and signed an information sheet providing their informed consent. Concealed allocation was performed by a blinded researcher, with block randomization for all patients and both groups, using the statistical program GraphPad version 8.0 (GraphPad Software, Inc., La Jolla, CA, USA) before the study started. Only the therapist had access to the allocation schedule. Both groups received eight sessions of 20 min for four weeks, twice a week. The procedure was performed according to previous studies [2,16]. 

DT equipment (CET 400 VA–RET 130 Watts–Winback, Biarritz, France), with a frequency of 500 kHz and intensity at 40%, was used. It was applied utilizing a flat capacitive head 4 cm in diameter. It was applied using longitudinal and transverse movements and a conductive cream for DT. DT was applied over the TP of the right and left trochanteric prominence based on evidence indicating it to be the most painful area in patients with FM [22]. The settings were calibrated so that patients felt no more than mild warmth during the treatment. This same intervention was simulated as a placebo in the control group; the DT was not activated while being administered. 

The treatment was carried out in a separate physiotherapy room in which the patient lay prone on a stretcher. The room was illuminated with fluorescent lamps, with no heat-generating electrical equipment and no incidence of sunlight or airflow on the participants. Room humidity and temperature were controlled using an electronic thermohygrometer (TFA-Dostmann, TFA 30.5045.54, Wertheim-Reicholzheim, Germany).

To successfully conceal pertinent information about the groups and not affect the ability to accurately assess or adjudicate results, single-blind masking was designed, and the control group was offered the possibility of intervention after the study was completed [41].

After the first intervention session, sociodemographic data were collected on a form in Google Forms, in which variables were collected such as age, weight, height, Body Mass Index (BMI), marital status, occupation, education, the year of diagnosis of FM, diagnosis of chronic fatigue, and several validated questionnaires were completed.

The pressure pain threshold (PPT) measurements were assessed using a digital pressure algometer (Fischer, analog model FPK60), with the pressure gradually increasing at a rate of 1 kg/s until they reported feeling pain and said “stop.” The mean value from three trials was calculated for analysis, with a 30 s resting period between each recording [30]. Measurements were conducted three separate times: before the intervention to establish baseline data, after eight sessions as a post-intervention measure, and 15 days after completing treatment.

### 2.4. Variables 

The Visual Analog Scale (VAS) assesses the overall pain experienced by the patient on a scale of 1–10, with higher scores indicating more pain. The VAS has proven to be a crucial tool in assessing pain and exhibiting sensitivity and specificity in evaluating pain in FMS [42].

Participants were assessed for their PPT using an algometer of the right and left trochanteric prominence.

The Fibromyalgia Impact Questionnaire (FIQ) measures the impact of FM. The values range between 0 and 100, with a higher score indicating a greater impact of the disease [43].

The Hospital Anxiety and Depression Scale (HADS) was used to measure anxiety and depression. The questionnaire consists of 14 items, divided into two subscales with 7 items each, using a Likert scale from 0 to 3. The odd-numbered items pertain to HADA, and the even-numbered ones relate to HADD. Each scale has a score range of 0–21 points. Higher scores indicate higher levels of anxiety and depression. Scores exceeding 11 are classified as “cases,” while those surpassing 8 are regarded as “probable cases” of anxiety and depression [44].

The Pittsburgh Sleep Quality Questionnaire (PSQI) was used to measure sleep quality. It is an efficient instrument for epidemiological and clinical research on sleep disorders. It assesses the following seven dimensions: sleep quality, delay in falling asleep, duration of sleep, perceived effectiveness of sleep, disturbances during sleep, use of medication for sleeping, and daytime dysfunction. Each aspect is assigned a score from 0 to 3. Lower scores indicate no challenges in the specific areas, while higher scores up to 3 indicate severe difficulties. The maximum achievable total score is 21, denoting critical sleep issues. A higher point total corresponds to a greater severity of sleep disorders [45].

The modified Fatigue Impact Scale (MFI-S) was used to measure self-reported general fatigue, which is a prominent FM symptom. A higher score indicated a more severe condition [46].

### 2.5. Statistical Analysis 

Data were analyzed using the IBM Statistics Package for Social Science, v.25 (IBM Corporation, Armonk, NY, USA). The Shapiro–Wilk test was used to analyze the normality of the variables (*p* > 0.05). Categorical data were described using frequency analyses. Baseline demographics and outcomes were compared between groups using independent Student’s *t*-tests for continuous data and chi-squared tests of independence for categorical data. The qualitative variables are expressed as frequencies (n (%)), and the quantitative variables as the mean (M) and standard deviation (SD). The analysis of variance (ANOVA) was used as a statistical analysis technique to compare the means of the intergroup and intragroup differences.

Given the existence of significant differences, the repeated measures were subsequently analyzed using the post hoc test with Bonferroni adjustment to further delve into the existing differences. A confidence level of 95% (*p* < 0.05) was used.

## 3. Results

### 3.1. Descriptive Study

A total of 31 patients with FM participated in the study, all of which were women. The mean age was 53.67 ± 8.13 years in the experimental group and 48.75 ± 10.9 years in the control group. The BMI was 28.18 ± 4.43 kg/m^2^ in the experimental group and 25.50 ± 5.61 kg/m^2^ in the control group. Overall, the socio-demographic characteristics of the sample were as follows: 18.5% had primary studies, 29.6% completed their Baccalaureate, 25.9% had professional education, and 25.9% had university studies. Only 29.6% were actively working. A total of 33.3% did not report data on their education, 37.0% reported lower levels, and 29.6% had medium levels. In total, 74.1% of the patients were married, 11.1% divorced and 14.8% single. Finally, 66.7% reported daily exercise (Table 1). For the descriptive study of the rest of the variables studied, including PPT, pain, disease impact, anxiety and depression, sleep quality, and fatigue, the sample presents similar baseline levels. 

### 3.2. Study of Normality

A non-normal distribution was found in the PPT variables of the right and left trochanter and the sleep quality variable with *p* < 0.05 (*p* = 0.001 and *p* = 0.011, respectively). Conversely, a normal distribution was observed for the pain variable, disease impact, anxiety, depression, and fatigue with *p* > 0.05 (*p* = 0.192, *p* = 0.200, *p* = 0.076, *p* = 0.200, and *p* = 0.200, respectively).

### 3.3. Statistic Analysis

Table 2 shows the intergroup differences in the means of the quantitative variables, resulting from the comparison of the measurements collected at the baseline, post-intervention, and response in fifteen days. Both groups started from similar baseline values. After the intervention, the measurements of both groups became more distant from each other. In response to fifteen days of measurements, the values continued to diverge with a tendency to favor the experimental group.

Table 3 details the intragroup differences throughout the investigation. In the experimental group, the PPT of the right trochanter in the post-intervention increased by 0.61, while in the control group, it decreased by 0.02. The PPT of the left trochanter in the experimental group increased by 0.88; in the control group, it increased by 0.03. A decrease in VAS of 0.94 was also observed in the experimental group, and in the control group, it increased by 0.08. 

An observed reduction in the disease impact was noted in the experimental group, with a decrease of 8.22 in the FIQ score, compared to only a 2.06 decrease in the control group. The experimental group experienced a reduction in sleep quality of 1.33, while the control group’s reduction was 0.67. The data collected on self-reported fatigue showed a decrease of 7.47 in the experimental group and 2.17 in the control group, indicating that participants in the experimental group experienced lower levels of fatigue at the beginning of the study. The results did not reveal any significant differences for the following pre-intervention quantitative variables between groups: PPT of the right (F = 1.72, *p* = 0.195) and left trochanters (F = 1.45, *p* = 0.849), pain (F = 2.27, *p* = 0.129), disease impact (F = 12.83; *p* = 0.209), anxiety (F = 1.33; *p* = 0.219), depression (F = 0.163; *p* = 0.688), sleep quality (F = 1.567; *p* = 0.024) and fatigue (F = 5.577; *p* = 0.022). The repeated measured ANOVA revealed significant group differences for the following post-intervention quantitative variables between groups: PPT of the right (F = 12.10; *p* < 0.001) and left trochanters (F = 1.45, *p* < 0.001), pain (F = 22.27, *p* < 0.001), disease impact (F = 12.83; *p* = 0.003), anxiety (F = 18.33; *p* < 0.001), depression (F = 21.34; *p* < 0.001), sleep quality (F = 17.96; *p* < 0.001) and fatigue (F = 5.577; *p* = 0.002).

The response in fifteen days for PPT values of the right trochanter increased in the experimental group values, allowing a better tolerance to pain. The disease impact decreased, anxiety was lower, and sleep quality improved. In contrast, the control group showed similar values during the response in fifteen days of intervention (Table 4).

The intergroup analysis was carried out in both groups using the post hoc statistic with Bonferroni adjustment. Statistically significant results were obtained in the PPT of the right trochanter only for the measurement that ranged from the pre-intervention to the response in fifteen days measurement (F = 1.45, *p* = 0.004) and the PPT of the left trochanter in all the moments of registration of the variables. Pre-intervention (F = 22.21, *p* = 0.0001), post-intervention (F = 26.80, *p* = 0.000) and response were measured over fifteen days (F = 20.85, *p* = 0.003).

Among the intragroup findings of the experimental group, there were statistically significant results in the post-intervention and response over fifteen days in the PPT values of the right trochanter for pre-intervention (F = 11.54, *p* = 0.008), post-intervention (F = 13.45, *p* = 0.002) and in the response over fifteen days (F = 18.72, *p* = 0.000), and the left trochanter as follows: pre-intervention (F = 21.31, *p* = 0.000), post-intervention (F = 18.19, *p* = 0.000), and in the response in fifteen days (F = 15.14, *p* = 0.000). No statistically significant results were detected for pain, disease impact, depression, or sleep quality. Statistically significant results were obtained for anxiety (F = 1.11, *p* = 0.048) as well as for fatigue (F = 0.79, *p* = 0.049). 

The results of the post hoc statistics in the control group did not show statistically significant results.

Figure 1 shows an increase in the PPT of the right trochanter in the experimental group after the intervention and 15 days later, while a decrease was observed in the control group.

The PPT in the left trochanter increased substantially in the experimental group compared to the control group.

## 4. Discussion

This study aimed to assess if the use of DT in individuals with FM yields notably beneficial outcomes regarding alleviating pain, as assessed through VAS and algometry.

The effect that DT can have on patients suffering from chronic pain is to reduce pain since by increasing the temperature of the tissues, blood circulation improves, allowing for recovery. 

In this study, as reflected in the main objective, we intended to analyze whether DT is an effective tool to reduce pain in patients with FM and alleviate the rest of the symptoms. 

According to the algometry (PPT) and VAS, the results of measurements taken during the pre-intervention and response in fifteen days were compared between the two groups, including both the experimental and control groups.

The experimental group was exposed to electromagnetic signals during eight sessions of DT (with a frequency of 500 kHz and intensity at 40%), with two sessions per week over 4 weeks. In contrast, the control group underwent a simulated intervention. The decision on the number of sessions utilized in this study was informed by prior research. [3]. 

The most important results were obtained for the PPT, anxiety, and fatigue concerning the group that received DT. However, for the variables of pain, the impact of the disease, depression, and sleep quality, no statistically significant results were obtained. The experimental design of this study was based on research conducted by Ibañez Vera, which aimed to study the impact of monopolar dielectric radiofrequency on FM symptoms [3]. Their sample consisted of 66 patients distributed into three groups, one experimental, one placebo, and one control, with a mean age of 47 ± 17.7. The sample members received eight sessions of 20 min after reporting frequent pain in that area in a prior assessment. In our study, the DT was applied to the trochanter since a later study specifically showed that a TP presenting more pain in FM corresponds to the trochanteric prominence [29].

Another difference between our research and previous studies is that we analyzed the consequences of DT 15 days following the intervention to evaluate its ongoing effectiveness. In the experimental group in this study, DT seemed to have a significant influence on the decrease in pain according to algometry in both trochanteric prominences. In contrast, pain decreased but was not statistically significant. Anxiety and fatigue values decreased after the interventions, but there were no statistically significant results in the response over fifteen days. The disease impact was reduced by 9.77 at the end of the investigation; although this was not significant, it is a point to highlight. The research indicates that the improvement in the sleep quality index for the patients was found to be significantly lower compared to both their initial level and the control group, suggesting a positive impact. The control group did not yield any statistically significant results, suggesting that DT treatment may have positive effects on the variables assessed in this study.

Ibañez Vera et al. [3] analyzed the effects of monopolar dielectric radiofrequency on the symptoms of FM and found that, in the short term, the therapy relieved pain and improved quality of life-related to the physical factor in participants with FM, but no significant between-group differences were found concerning the quality of life. To our knowledge, this study is one of the few that aims to verify whether DT is effective on pain in patients with FM. Above all, it is the only one that measures respond for the effects over fifteen days.

Based on the results obtained, DT can be said to be an effective tool to reduce pain in the short term and response in over fifteen days in patients with FM.

Babaei-Ghazani et al. explored the effectiveness of shortwave DT on pain, function, and grip strength in patients with chronic lateral epicondylitis, also using sham DT, and both groups showed a significant decrease in pain and an improvement in function [47]. Patients can also experience positive effects from placebos, sham treatments, or contextual factors.

Unlike other electrotherapy instruments, DT allows treatment to be carried out on patients with some type of metal prosthesis since it does not transmit heat to the metal material, which undoubtedly provides a new tool in the fight against chronic pain [48]. 

Transcutaneous electrical nerve stimulation (TENS) is one of the most studied and used non-invasive treatments in FM. In a systematic review carried out by Megía-García [49], they concluded that TENS is effective in reducing pain in people with FM, but efficacy in variables other than pain was not demonstrated. However, in the current study, the value of other variables, such as disease impact, anxiety, sleep quality, and fatigue, seemed to be improved.

Contrary to the above, it appears that research indicates that TENS may not be highly effective for individuals with FM, as it is better suited to treating specific areas of pain rather than the widespread and generalized pain characteristic of FM. This limits its effectiveness in managing the condition [50].

Moretti et al. analyzed the influence of receiving one or two weekly sessions of a combined ultrasound and interferential therapy in patients with FM. Although no significant differences were found between the groups, both treatments led to improvements in quality of life, sleep quality, and musculoskeletal symptoms. The researchers assessed pain using VAS for seven days after treatment and found notable enhancements in both groups [51].

In any case, the cause and pathogenesis of fibromyalgia are not yet clarified, so there is no standard therapy or clinical guide for its management. The effects of various non-pharmacological treatments have been studied in patients with fibromyalgia, including aerobic exercise, resistance exercise, chiropractic, Pilates, yoga, and balneotherapy, apart from those previously mentioned [52,53,54,55,56,57].

Acupuncture is a tool used to reduce pain in FM. There was a randomized clinical trial [58] in which the effects of acupuncture were studied in 100 patients. The study lasted 12 weeks, with a frequency of 2 sessions of 30 min per week, and no significant differences were observed in pain, fatigue, sleep, or general well-being. In contrast, a study on electroacupuncture [59], in which 36 patients with FM participated, receiving six sessions over 3 weeks, showed improvements in PPT, pain scales, and quality of sleep but not in morning stiffness and regional pain. The main difference in our study is that DT is not considered invasive, as in the case of acupuncture or electroacupuncture.

Finally, some potential limitations are acknowledged. First, our study only included women with FM due to the limited availability of male participants, so our findings may not be generalizable to male individuals with FM. Second, although we assessed depression and anxiety to analyze the contribution of psychological variables to CS, further research should consider analyzing the contribution of other factors, such as catastrophism or pain hypervigilance, to CS in this pain population. Finally, our sample size was small, and, therefore, our results should be considered preliminary normative data; further research with a larger sample size is needed to confirm these findings.

## 5. Conclusions

DT has been shown to increase PPT values of the right trochanter, allowing for better tolerance to pain in our patients for the obtained results in its evaluation, using the VAS measurement tools and algometry.

The results obtained show that this treatment can improve the quality of sleep and the impact of disease, chronic fatigue, and anxiety in patients with fibromyalgia.

No results have been observed in terms of depression and the impact of disease in patients with fibromyalgia.

## Figures and Tables

**Figure 1 biomedicines-12-01465-f001:**
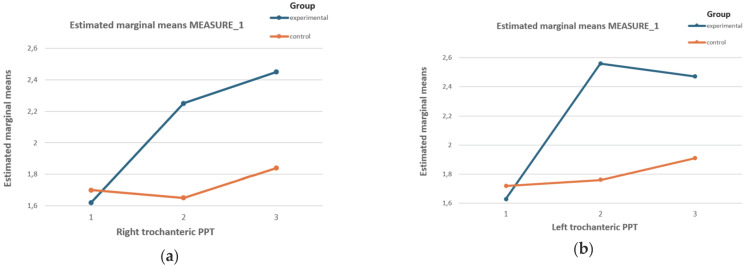
Effects of the DT on the PPT values of (**a**) the right and (**b**) the left trochanter of both groups for the three measurement stages. Graph (**a**) of the effects of the DT on the PPT of the right trochanter of both groups in the three measurement stages. Graph (**b**) for the effects of DT on the PPT of the left trochanter of both groups in the three measurement stages.

**Table 1 biomedicines-12-01465-t001:** Sample sociodemographic characteristics.

	Sample Characteristics (*n* = 27)	Experimental Group (*n* = 15)	Control Group (*n* = 12)
Age (years) (M ± DT)	(51.48 ± 9.62)	(53.67 ± 8.13)	(48.75 ± 10.9)
BMI (M ± DT)	(26.99 ± 5.07)	(28.18 ± 4.43)	(25.5 ± 5.61)
Level of education N (%)			
Primaria	5 (18.5%)	2 (13.3%)	3 (25%)
Baccalaureate	8 (29.6%)	5 (33.3%)	3 (25%)
Professional education	7 (25.9%)	4 (26.7%)	3 (25%)
University studies	7 (25.9%)	4 (26.7%)	3 (25%)
Employment status N (%)			
Unemployed	14 (51.9%)	10 (66.7%)	4 (33.3%)
On medical leave	3 (11.1%)		3 (25%)
Actively working	8 (29.6%)	3 (20%)	5 (41.7%)
Pensioner	2 (7.4%)	2 (13.3%)	
Buying power N (%)			
NS/NR	9 (33.3%)	5 (33.3%)	4 (33.3%)
Lower levels	10 (37.0%)	5 (33.3%)	5 (41.7%)
Medium levels	8 (29.6%)	5 (33.3%)	3 (25%)
Marital status N (%)			
Single	4 (14.8%)	1 (6.7%)	3 (25%)
Divorced	3 (11.1%)	2 (13.3%)	1 (8.3%)
Married	20 (74.1%)	12 (80%)	8 (66.7%)
Exercise			
3 o + times a week	3 (3.7%)	3 (20%)	8 (66.7%)
1–3 times a week	17 (63%)	9 (60%)	
Never	7 (25.9%)	3 (20%)	4 (33.3%)

Note: Values of quantitative variables are expressed as mean (M) and standard deviation (SD) and values of qualitative variables as frequencies and percentages N (%).

**Table 2 biomedicines-12-01465-t002:** Intergroup differences in the means of the quantitative variables: right and left trochanter pressure pain threshold, pain, disease impact, anxiety and depression, sleep quality, and fatigue.

	Pre-Intervention				Post-Intervention				Response in Fifteen Days			
	Diathermy ON	Diathermy OFF	Mean Differences	*p*-Value	Diathermy ON	Diathermy OFF	Mean Differences	*p*-Value	Diathermy ON	Diathermy OFF	Mean Differences	*p*-Value
PPT right trochanter	1.64 ± 0.61	1.68 ± 1.03	0.04	0.093	2.25 ± 0.58	1.66 ± 0.59	0.59	0.004 *	2.45 ± 0.53	1.86 ± 0.65	0.59	0.063
PPT left trochanter	1.65 ± 0.48	1.70 ± 0.73	0.05	0.001 *	2.53 ± 0.46	1.73 ± 0.65	0.80	0.000 *	2.48 ± 0.73	1.90 ± 0.75	0.58	0.003 *
Pain (0–10)	6.07 ± 2.05	7.25 ± 1.60	1.18	0.11	5.13 ± 1.80	7.33 ± 1.61	2.20	0.45	5.87 ± 1.50	6.42 ± 1.56	0.55	0.37
Disease impact (0–100)	63.70 ± 12.02	74.44 ± 9.65	10.74	0.38	55.48 ± 17.20	72.38 ± 10.09	16.9	0.51	53.93 ± 16.46	68.61 ± 11.65	14.68	0.38
Anxiety (HADS)	10.40 ± 2.58	12.67 ± 2.64	2.27	0.62	10.33 ± 2.76	12.25 ± 2.63	1.92	0.40	9.33 ± 2.41	11.83 ± 2.03	2.50	0.11
Depression (HADS)	12.67 ± 4.70	7.17 ± 3.90	5.50	0.70	13.40 ± 4.30	7.42 ± 5.29	5.98	0.51	13.67 ± 3.95	7.42 ± 4.88	6.25	0.51
Sleep quality (0–21)	15.60 ± 3.58	17.00 ± 1.95	1.40	0.58	14.27 ± 3.63	16.33 ± 2.70	2.06	0.23	14.07 ± 3.53	16.75 ± 2.18	2.68	0.14
Fatigue (MFI-S)	75.67 ± 12.07	83.17 ± 8.33	7.50	0.28	68.20 ± 15.61	81.00 ± 8.90	12.8	0.29	69.00 ± 14.41	81.75 ± 13.87	12.75	0.94

PPT: pressure pain threshold; HADS: Hospital Anxiety and Depression Scale; MFI-S: modified Fatigue Impact Scale. * *p* ≤ 0.05.

**Table 3 biomedicines-12-01465-t003:** Intragroup differences in the means of the quantitative variables: right and left trochanteric PPT, pain, disease impact, anxiety and depression, sleep quality, and fatigue.

	Difference Pre-Post				Difference Post-Response in Fifteen Days				Difference Pre-Response in Fifteen Days			
	Diathermy ON	*p*-Value	Diathermy OFF	*p*-Value	Diathermy ON	*p*-Value	Diathermy OFF	*p*-Value	Diathermy ON	*p*-Value	Diathermy OFF	*p*-Value
PPT right trochanter	0.61	0.008 *	−0.02	0.941	0.20	0.002 *	0.20	0.452	0.81	0.000 *	0.18	0.652
PPT left trochanter	0.88	0.000 *	0.03	0.872	−0.05	0.000 *	0.17	0.184	0.83	0.000 *	0.20	0.480
Pain (0–10)	−0.94	0.074	0.08	0.809	0.74	0.748	−0.91	0.127	−0.20	0.252	−0.83	0.127
Disease impact (0–100)	−8.22	0.167	−2.06	0.493	−1.55	0.055	−3.77	0.071	−9.77	0.120	−5.83	0.106
Anxiety (HADS)	−0.07	0.915	−0.42	0.601	−1.00	0.048 *	−0.42	0.301	−1.07	0.112	−0.84	0.552
Depression (HADS)	0.73	0.419	0.25	0.777	0.27	0.196	0.00	0.775	1.00	0.442	0.25	0.930
Sleep quality (0–21)	−1.33	0.073	−0.87	0.523	0.20	0.091	0.42	0.612	−1.53	0.096	−0.25	0.692
Fatigue	−7.47	0.071	−2.17	0.389	0.80	0.030 *	0.75	0.740	6.67	0.049 *	−1.42	0.810

PPT: pressure pain threshold; HADS: Hospital Anxiety and Depression Scale; MFI-S: modified Fatigue Impact Scale. * *p* ≤ 0.05.

**Table 4 biomedicines-12-01465-t004:** Intergroup differences in the means of the quantitative variables between the baseline measurement moments and response in fifteen days.

	Diathermy ON (M ±DS)	Diathermy OFF (M ±DS)	Intergroup Mean Differences
PPT right trochanter	0.81 ± 0.80	0.18 ± 0.82	0.63
PPT left trochanter	0.83 ± 0.70	0.20 ± 0.49	0.62
Pain (0–10)	−0.20 ± 2.37	−0.83 ± 1.75	0.63
Disease impact (0–100)	−9.77 ± 18.07	−5.83 ± 10.09	3.95
Anxiety (HADS)	−1.07 ± 1.91	−0.84 ± 2.66	0.24
Depression (HADS)	1.00 ± 2.85	0.25 ± 2.96	0.75
Sleep quality (0–21)	−1.53 ± 3.27	−0.25 ± 1.66	1.28
Fatigue	−6.67 ± 10.69	−1.42 ± 14.44	5.26

PPT: pressure pain threshold; HADS: Hospital Anxiety and Depression Scale; MFI-S: modified Fatigue Impact Scale.

## Data Availability

Data used for this manuscript are available upon request.

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
