# Peer review of "Effects of Diathermy on Pain in Women with Fibromyalgia: A Randomized Controlled Trial"

_biomedicines, 2024, doi:10.3390/biomedicines12071465_

Round 1

Reviewer 1 Report

Comments and Suggestions for Authors

The present study investigated the effect of diathermy on some symptoms found in patients with fibromyalgia. For the manuscript to be accepted, authors must meet and respond to the requests below:

In addition to approval from the ethics committee, was the study not previously registered in a registry database, such as trial.com?

How was randomization carried out? Please describe in the text;

Although it was based on a previous study, a sample size calculation was not performed?

It is necessary to describe what the DT application room was like. Was there a Faraday cage in the room?

Was there a sample loss at the end of the study? If yes, was an intention-to-treat analysis performed? If not, justify.

In results, it is interesting to present the p values ​​for all tables in which the analyzes were carried out. This will make it easier for readers to understand.

Why were non-parametric data not presented as medians?

It is important to present and compare the baseline, post-intervention and follow-up data in the same table with the p-values ​​or the most appropriate confidence interval.

There are already some works in the literature, such as Masiero et al. (doi: 10.1007/s00484-019-01806-x), who evaluated the effect of diathermy on fibromyalgia, including some forms similar to the one in the present study. What would be the authors' justification for the originality of this work?

Author Response

Reviewer Comments to Author:

Reviewer 1:

Comment 1:

In addition to approval from the ethics committee, was the study not previously registered in a registry database, such as trial.com?

The authors appreciate this comment. You are right. The study was approved by the Ethics Committee of the University Camilo José Cela, unfortunately, the authors did not register the study as a clinical trial.

Comment 2:

How was randomization carried out? Please describe in the text.

Thank you for your comments and the time devoted to correcting our study. We have added: “Concealed allocation was performed by a blinded researcher, with block randomization for all patients and both groups, using the statistical program GraphPad version 8.0 (GraphPad Software, Inc. La Jolla, CA) before the study started. Only the therapist had access to the allocation schedule.” (Line 130-133).

Comment 3:

Although it was based on a previous study, a sample size calculation was not performed?

Thank you for pointing this out. The sample size for this work was calculated based on our previous study, in which the sample size corresponded to a population of 30 patients.

Comment 4:

It is necessary to describe what the DT application room was like. Was there a Faraday cage in the room?

Thanks for this comment. You are right. Following your comment, we have added the description of the room: “The treatment was carried out in a separate physiotherapy room, in which the patient lay prone on a stretcher. The room was illuminated with fluorescent lamps, with no heat-generating electrical equipment and no incidence of sunlight or air-flow on the participants. Room humidity and temperature were controlled using an electronic thermo-hygrometer (TFA-Dostmann, TFA 30.5045.54, Germany).” (Lines 144-148).

In answer to your question, the physical therapy room was not equipped with a Faraday cage. The room was adapted for the study, and different prevention measures were considered, such as maintenance of the device, according to its manufacturer, to avoid leaks.

In contrast to the generalization that Faraday cages protect from exposure to oscillating electric and magnetic fields, in the study by Messias Ide A. et al. (2011) it was observed that the Faraday cage offers physical therapists no protection, and instead, increases their level of exposure.

  • Messias Ide A, Okuno E, Colacioppo S. Exposição ocupacional de fisioterapeutas aos campos elétrico e magnético e a eficácia das gaiolas de Faraday [Occupational exposure of physical therapists to electric and magnetic fields and the efficacy of Faraday cages]. Rev Panam Salud Publica. 2011 Oct;30(4):309-16. Portuguese. PMID: 22124689.

Comment 5:

Was there a sample loss at the end of the study? If yes, was an intention-to-treat analysis performed? If not, justify.

Thank you for pointing this out. In our research and during the follow-up period, three participants were no longer part of the control group - one withdrew by personal choice, another due to personal issues, and one woman due to COVID-19. In the experimental group, one participant dropped out for health-related reasons.

The individuals no longer included in the study were removed, and an intention-to-treat (ITT) analysis was not conducted. The data were only analyzed for those who finished the study: 12 participants in the control group and 15 participants in the experimental group.

The ITT analysis was not performed because we wanted to know what the effect was of assigning a treatment. As the study was designed, the average causal effect of receiving treatment was more important than ITT. The ITT analysis could have been skewed away from the null value since adherence rates could be different between groups. The proportion of adherence is related to the outcome, and our control group was a placebo group.

  • Shrier, I., Verhagen, E., Stovitz, S.D., 2017. The Intention-to-Treat Analysis Is Not Always the Conservative Approach. The American Journal of Medicine 130, 867–871.. https://doi.org/10.1016/j.amjmed.2017.03.023

Comment 6:

In results, it is interesting to present the p values ​​for all tables in which the analyzes were carried out. This will make it easier for readers to understand.

Thanks for this comment. You are right. We have included the p-values for all the tables as the reviewer suggested.

Comment 7:

Why were non-parametric data not presented as medians?

Thanks for this comment. Since practically all the variables analyzed followed a normal distribution, it was decided not to present the medians of the non-normal variables.

Comment 8:

It is important to present and compare the baseline, post-intervention and follow-up data in the same table with the p-values ​​or the most appropriate confidence interval.

The authors appreciate this comment. Thanks for this comment. You are right. Following your comment, we have added the p-values in the tables.

Reviewer 2 Report

Comments and Suggestions for Authors

The aim of the presented paper was to analyze the efficacy of diathermy on pain in patients with fibromyalgia. The results, which are quite promising, highlighted that this treatment has significantly improved the quality of sleep, the impact of disease, chronic fatigue, and anxiety in patients with fibromyalgia. This could potentially lead to a breakthrough in fibromyalgia treatment. 

The manuscript is well structured, with the main purpose so clear. 

However, I have important suggestions that, if implemented, could significantly enhance the quality and impact of this research. 

1. Procedure: Please add data regarding the division into control and treatment groups in your work. There is no data regarding, for example, age in individual subgroups. As the age range of the surveyed women was between 30 and 70 years old, this could have impacted the experiment results.

 2. The discussion makes few references to the literature regarding other fibromyalgia treatment methods and results. Only a few studies are mentioned, and the topic requires attention in the context of the authors' results.

OVERALL:

In general, the review is interesting and can contribute to the literature. I hope my suggestions will help improve this work.

Author Response

Reviewer Comments to Author:

Reviewer 2:

The presented paper aimed to analyze the efficacy of diathermy on pain in patients with fibromyalgia. The results, which are quite promising, highlighted that this treatment has significantly improved the quality of sleep, and the impact of disease, chronic fatigue, and anxiety in patients with fibromyalgia. This could potentially lead to a breakthrough in fibromyalgia treatment.

The manuscript is well structured, with the main purpose so clear.

However, I have important suggestions that, if implemented, could significantly enhance the quality and impact of this research.

Comment 1:

  1. Procedure: Please add data regarding the division into control and treatment groups in your work. There is no data regarding, for example, age in individual subgroups. As the age range of the surveyed women was between 30 and 70 years old, this could have impacted the experiment results.

The authors appreciate this comment. Thanks for this comment. Age characteristics were similar in both groups. There were no statistically significant differences between the groups. The age ranges between the groups showed similar percentages to compare the results.

Comment 2:

  1. The discussion makes few references to the literature regarding other fibromyalgia treatment methods and results. Only a few studies are mentioned, and the topic requires attention in the context of the authors' results.

Thanks for this comment. You are right. Following your comment, we have added the following text: “Moretti et al. (2012) analyzed the influence of receiving one or two weekly sessions of combined ultrasound and interferential therapy in patients with FM. Although no significant differences were found between the groups, both treatments led to improvements in quality of life, sleep quality, and musculoskeletal symptoms. The researchers assessed pain using VAS seven days after treatment and found notable enhancements in both groups.

In any case, the cause and pathogenesis of fibromyalgia is not yet clarified, so there is no standard therapy or clinical guide for its management. The effect of various non-pharmacological treatments has been studied in patients with fibromyalgia, including aerobic exercise, resistance exercise, chiropractic, Pilates, yoga and balneotherapy, apart from those previously mentioned.” (Lines 165-175)

Please consider these references:

  • Moretti FA, Marcondes FB, Provenza JR, Fukuda TY, de Vasconcelos RA, Roizenblatt S. Combined therapy (ultrasound and interferential current) in patients with fibromyalgia: once or twice in a week? Physiother Res Int. 2012 Sep;17(3):142-9. doi: 10.1002/pri.525. Epub 2011 Nov 24. PMID: 22114059.
  • Hong-Baik I, Úbeda-D'Ocasar E, Cimadevilla-Fernández-Pola E, Jiménez-Díaz-Benito V, Hervás-Pérez JP. The Effects of Non-Pharmacological Interventions in Fibromyalgia: A Systematic Review and Metanalysis of Predominants Outcomes. Biomedicines. 2023 Aug 24;11(9):2367. doi: 10.3390/biomedicines11092367. PMID: 37760809; PMCID: PMC10525643.
  • Gunendi, J. Meray and S. Ozdem, The effect of a 4 week aerobic exercise program on muscle performance in patients with fibromyalgia, J Back Musculoskelet Rehabil 21 (2008), 185–191.
  • B. Panton, A. Figueroa, J.D. Kingsley, L. Hornbuckle, J. Wilson, N. St. John, D. Abood, R. Mathis, J. VanTassel and V. McMillan, Effects of Resistance Training and Chiropractic Treatment in Women with Fibromyalgia, J Altern Complement

Med 15(3) (2009), 321–328.

  • Altan, N. Korkmaz, ¨U. Bingol and B. Gunay, Effect of Pilates training on people with fibromyalgia syndrome: a pilot study, Arch Phys Med Rehabil 90 (2009), 1983–1988.
  • Gerson,M.S. daSilva, L.F.Geraldo, V. Lais andM.D. Lage, Effects of yoga and the addition of Tui Na in patients with fibromyalgia, J Altern Complement Med 13 (2007), 1107– 1113.
  • Evcik, B. Kizilay and E. G¨okc¸en, The effects of balneotherapy on fibromyalgia patients, Rheumatol Int 22 (2002), 56–59.

Reviewer 3 Report

Comments and Suggestions for Authors

Dear Authors,

I have read the manuscript and I send you my comments:

1) please add the power calculation

2) please add the clinical characteristics of the enrolled patients and the treatment for each of these

3) please add the statistical correlation between clinical effects and characteristics of the patients

4) the rime of the follow-up is very short please increase it

5) please add a control group

Comments on the Quality of English Language

none

Author Response

Reviewer Comments to Author:

Reviewer 3:

Comment 1:

1) please add the power calculation:

Thank you for your comments and the time devoted to correcting our study. Finally, it was decided not to analyze the power of the study since the sample size was 30 patients, which was considered a small sample size.

Comment 2:

2) please add the clinical characteristics of the enrolled patients and the treatment for each of these

The authors appreciate this comment. Thanks for this comment. In the results section we have described the clinical characteristics of the patients; we decided not to include the table so as not to saturate the document with duplicated information; however, if you consider that we should include the table, we have it done.

Comment 3:

3) please add the statistical correlation between clinical effects and characteristics of the patients

The correlation between the clinical factors and the characteristics of the patients was not carried out since it did not correspond to the main objective of the study, which was intended to analyze whether the application of diathermy managed to reduce pain.

Comment 4:

4) the rime of the follow-up is very short please increase it

Thank you for your comments and the time devoted to correcting our study. Some reviewers have not indicated that in order to consider the follow-up valid, it has to be a minimum period of 6 weeks up to 12 weeks, so we decided to evaluate acute responses up to 15 days, nevertheless, we consider their proposal of great value and we will include it in future research.

Comment 5:

5) please add a control group

The authors appreciate this comment. Thanks for this comment. The study already has a control group, which is described in the Materials and Methods section. If you consider that we need to add any information regarding the control group, please let us know so that we can make the appropriate contribution.

Round 2

Reviewer 1 Report

Comments and Suggestions for Authors

The authors met all the requests requested, which leaves the manuscript capable of being accepted for publication.

Author Response

Thank you for your comments and the time devoted to correcting our study.

Reviewer 3 Report

Comments and Suggestions for Authors

Dear Authors,

I have read the revised version of your manuscript, but you have not do the requested revisions

Comments on the Quality of English Language

none

Author Response

Reviewer Comments to Author:

Reviewer 3:

Comment 1:

1) please add the power calculation:

Thank you for your comments and the time devoted to correcting our study. Based on a previous study conducted by Ge et al., a minimum sample size of at least 30 patients with FMS could be considered as appropriate for this study.

  • Ge, H.-Y.; Wang, Y.; Danneskiold-Samsøe, B.; Graven-Nielsen, T.; Arendt-Nielsen, L. The Predetermined Sites of Examination for Tender Points in Fibromyalgia Syndrome Are Frequently Associated with Myofascial Trigger Points. J. Pain 2010, 11, 644–651.

Comment 2:

2) please add the clinical characteristics of the enrolled patients and the treatment for each of these

The authors appreciate this comment. Thanks for this comment. The sociodemographic characteristics of the population are described in lines 209 to 219; nevertheless, we add Table 1 with the results described.

Comment 3:

3) please add the statistical correlation between clinical effects and characteristics of the patients

The authors appreciate this comment. Unfortunately, the correlation between the clinical factors and the characteristics of the patients was not carried out because it did not correspond to the main objective of the study, which was to analyze whether the application of diathermy succeeded in reducing pain.

Comment 4:

4) the rime of the follow-up is very short please increase it

Thank you for your comments and the time devoted to correcting our study. Some reviewers have not indicated that to consider the follow-up valid. Following your terminology, we have changed the term follow-up to RESPONSE IN FIFTEEN DAYS.

Comment 5:

5) please add a control group

The authors appreciate this comment. Thanks for this comment. The study already has a control group, which is described in the Materials and Methods section (“An experimental study design was carried out through a single-blind, randomized study. Patients were randomly assigned to either the experimental or the control group”). The experimental group during the study was named “Diathermy ON” and the control group was named “Diathermy OFF”.

After making all the modifications indicated by the reviewers, the document has been revised again by a native English speaker, Mary Fiona Mc Fall (mmcfall@ucjc.edu) Bachelor in English Philology, Master in Secondary Education, Vocational Training and EOI (Specialty Modern Languages - English), Master in Applied English Linguistics: (Specialty Sociolinguistics). 
